# Video meeting signals: Experimental evidence for a technique to improve the experience of video conferencing

**Paul D. Hills**[1], **Mackenzie V. Q. Clavin**[1], **Miles R. A. Tufft**[1], **Matthias S. Gobel**[2], **Daniel C. Richardson**[1]*

1 Department of Experimental Psychology, University College London, London, United Kingdom,
2 Department of Psychology, University of Exeter, Exeter, United Kingdom

* dcr@eyethink.org

**Data Availability Statement:** Data are freely available at https://osf.io/jzm28/.

**Funding:** MSG received support to run Experiment 2 from the University of Exeter Inclusivity Project

## Abstract

We found evidence from two experiments that a simple set of gestural techniques can improve the experience of online meetings. Video conferencing technology has practical benefits, but psychological costs. It has allowed industry, education and social interactions to continue in some form during the covid-19 lockdowns. But it has left many users feeling fatigued and socially isolated, perhaps because the limitations of video conferencing disrupt users' ability to coordinate interactions and foster social affiliation. Video Meeting Signals (VMS™) is a simple technique that uses gestures to overcome some of these limitations. First, we carried out a randomised controlled trial with over 100 students, in which half underwent a short training session in VMS. All participants rated their subjective experience of two weekly seminars, and transcripts were objectively coded for the valence of language used. Compared to controls, students with VMS training rated their personal experience, their feelings toward their seminar group, and their perceived learning outcomes as significantly higher. Also, they were more likely to use positive language and less likely to use negative language. A second, larger experiment replicated the first, and added a condition where groups were given a version of the VMS training but taught to use emoji response buttons rather than gestures to signal the same information. The emoji-trained groups did not experience the same improvement as the VMS groups. By exploiting the specific benefits of gestural communication, VMS has great potential to overcome the psychological problems of group video meetings.

## Introduction

People struggle with communication technologies when they clash with social norms. Users of the newly invented telephone, in the early 19th century, did not know how to greet each other. They couldn't see the other person, did know their age, gender or standing, and so how could they know what greeting was appropriate? The solution, urged by Thomas Edison, was to adopt the neutral word 'hello', which until that point was an uncommon word used to express surprise. In this way, social customs adapted to new technology.

(https://theinclusivityproject.co.uk) which is
supported by the European Regional Development
Fund (https://ec.europa.eu/regional_policy/en/
funding/erdf/). The funders had no role in study
design, data collection and analysis, decision to
publish, or preparation of the manuscript.

**Competing interests:** PDH developed Video
Meeting Signals (VMS) and markets workshops
and training on this technique. This competing
interest does not alter the authors' adherence to
PLOS ONE policies on sharing data and materials.

Video conferencing, at first glance, appears to be much more like face-to-face interaction than the telephone. However—as millions of people have recently discovered [1]—the advantages of video conferencing can come at a cost. The use of computer mediated communication during the pandemic lockdown was associated with poor mental well-being [2], and resulted in communication across organizations becoming more 'siloed and stilted' [3]. In a study of over a thousand video conferences, it was found turning the camera on (compared to leaving it off) caused feelings of daily fatigue that impacted engagement and could last until the following day [4].

One speculation is that video conferencing produces fatigue because it lacks or disrupts many of the nuances of real life social interaction [5]. The crucial role that eye contact plays in social interaction [6–9] is impossible with dislocated screens and cameras. Gaze shifts [10–12] from one person to another cannot be followed. The rhythm of conversation is disrupted over zoom, with one study finding that transition times between speakers increased threefold during video conferences [13]. And small video windows and muted mics cannot convey all the backchanneling behaviour of nods, murmurs and gestures that are vital to the orchestration of spontaneous conversation [14–17] and the building of rapport [18].

Some new technologies and tweaks have emerged in response to these problems. For example, both Zoom and Microsoft Teams have introduced response buttons that can be used to signal encouragement or approval via an emoji superimposed on the users' video. In this paper, we compare those systems with a different approach, that follows Edison in adopting and adapting a pre-existing behaviour to the new technology. It uses a communication technique that possibly predates spoken language: gesture [19].

One of us (P.D.H.) developed a system of Video Meeting Signals (VMS) that can be easily taught and employed during video conferences [20]. The technique was developed over time while working with online groups, resulting in a set of gestures that were readily adapted, easily remembered and used spontaneously. The signals have easily interpretable, iconic meanings—such as a wave to attract attention, or thumbs-up for approval—and are designed to be easily visible (Fig 1). In this way, the signals have the same function as subtle backchannel, communicative behaviours that might be used face to face—such as nods, smiles or raised eyebrows—but are visible even in a small video window. In two experiments, we explored whether and why this technique could improve the experience of video conferencing.

## Experiment 1

In our first experiment, we carried out a randomised controlled trial to test the efficacy of VMS with first year undergraduate students at UCL. The students have weekly seminar meetings throughout the term, led by a faculty member. In the 2020–21 school year, all teaching at UCL was conducted online. 14 seminar groups consented to be in the trial in the second term of the year. Half of these were randomly assigned to the experimental condition and received a training session in VMS. For the following two weeks, students filled in surveys after their seminars, evaluating their personal experience and their feelings of affiliation towards their seminar group. They also rated the mechanics of their interaction (how the class was structured and how easy it was to speak up), and learning outcomes (the progress they made on their projects or if they had learned from discussing a paper). In addition, 8 of the seminar groups consented to their seminars being recorded and automatically transcribed by the Zoom software. The anonymised transcript was then analysed for the emotional valence of each utterance, using the AWS comprehend natural language processor [21].

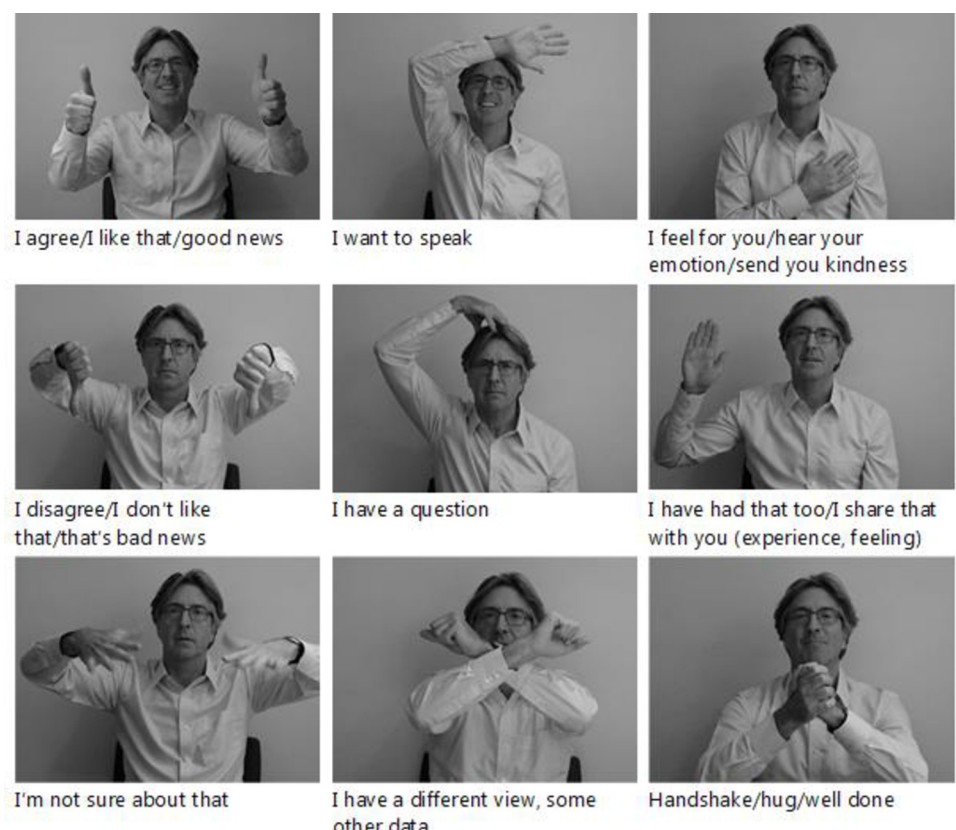

**Fig 1. Video meeting signals taught during the training session of Experiment 1 (demonstrated by P.D.H).**

## Methods

**Participants.** A total of 12 undergraduate seminar groups from year one of the UCL BSc Psychology course agreed to take part in exchange for course credit. Seminar groups were made up of one seminar leader (a member of UCL teaching staff) and approximately 10 students. The majority of participants in the groups (N = 99) were female. Since students had already been allocated into seminar groups, we used a cluster randomised design, and randomly assigned 6 groups to the experimental (N = 62) and 6 to the control (N = 56) conditions. A total of 109 participants in week 1, and 105 participants in week 2 filled in surveys following their seminars. Fully informed consent was obtained online, and participants' data were anonymised throughout.

79 participants in 8 groups additionally consented to their seminars being recorded so that audio transcripts could be generated for analysis. To ensure full anonymity of transcript data, students and seminar leaders were instructed to rename their Zoom name using a randomly allocated and unique noun codeword provided prior to each session. Ethical approval was provided prior to data collection by the UCL board (3828/003, EP/2021/006).

**Procedure.** Data collection was carried out across two one-hour seminar sessions in weeks 7 & 8 (referred to weeks 1 & 2 in analysis) of term 2 of the UCL BSc Psychology course. All sessions were conducted online using the Zoom platform. In the preceding week, experimental groups (students and seminar leaders) attended a 45 minute session with an experimenter trained in the VMS technique and were provided with detailed guidance material on

the VMS method. Conversely, control groups were kept blind of the method and did not receive any training or additional material.

Groups in both conditions were told to conduct both seminar sessions as planned, but the experimental groups were instructed to implement the VMS method as learnt. No other aspect of the seminar approach was changed, and the term's planned curriculum remained unchanged. The first week was a group discussion of hypotheses for an upcoming lab experiment, and the second week was a journal club discussion of an academic paper. All participants were instructed to have their cameras on during the seminar. The text chats function on zoom was typically not used during seminars, other than to occasionally share URLs, and at this point in time, the zoom platform had not implemented the use of response buttons. Therefore, participants in all groups communicated through spoken, or non-verbal means.

Immediately after each session participants were required to complete a post-seminar survey. Since we could not find a pre-exiting psychometric scale that was appropriate to our needs, we wrote out a set of questions asking about a range of intra and interpersonal factors, the experience of the social interaction, and the educational outcomes of the seminar. We then grouped these survey questions into four themes: whether they related to participants' *group affiliation* (e.g., 'I know my group members well'), the *learning outcomes* of that seminar (e.g., 'The seminar had a positive impact on my learning'), their *personal experience* (e.g., 'I enjoyed the seminar') and the *mechanics* of the seminar interaction (e.g. 'I found it hard to speak'). Importantly, since these were not preexisting, validated scales and the themes were subjective, we report means for each individual item below, and our full dataset is available online. S1 File contain a list of survey items, their themes, and indicate where items were reversed scored.

For the 8 groups that agreed to record their sessions, anonymised audio transcripts were generated automatically by the Zoom platform, uploaded to AWS cloud computing service, and analysed with the sentiment function of the comprehend package.

**Statistical analyses.** We used Bayesian mixed models to quantify the evidence that each of our experimental factors influenced participants' survey responses or transcripts. Mixed models are able to account for the fact that participants were nested in particular seminar groups, and the Bayesian approach avoids some of the problems associated with null hypothesis testing [22, 23].

Our mixed models used fixed effects for the training condition, the seminar session and the survey theme. There were random effects for the seminar group and the participant, with random intercepts. We used R [24] and the package rstanarm [25], employing weakly informative priors that were scaled following the standard rstanarm procedure (full priors are reported in the S1 File). From 4000 samples, we generated estimates of the posterior distributions of the model parameter coefficients, which quantify the strength of the evidence that each experimental condition influenced behaviour, using the package psycho [26].

Survey responses were reversed-scored where necessary and normalised within each item. Using the formula notation in the R stats package, the full model was specified as:

response ~ training * week * theme + (1 | seminar group) + (1 | participant)

For the transcript data, we used the same Bayesian mixed model analysis, specified as:

response ~ training * week * valence + (1 | seminar group) + (1 | participant)

S1 File give the explanatory power and full parameter estimates of each model with Median, Median Absolute Deviance (MAD), 95% Confidence-Interval (CI- CI+), Maximum Probability of Effect (MPE) and Overlap for each term.

**Results.** Fig 2 shows how survey responses and transcripts differed between experimental and control groups. The models provide estimates of the distribution of condition means, and the maximum probability of effect (MPE), which quantifies the probability that the condition means differ. An MPE of above 90% can be thought of as 'strong evidence' [27].

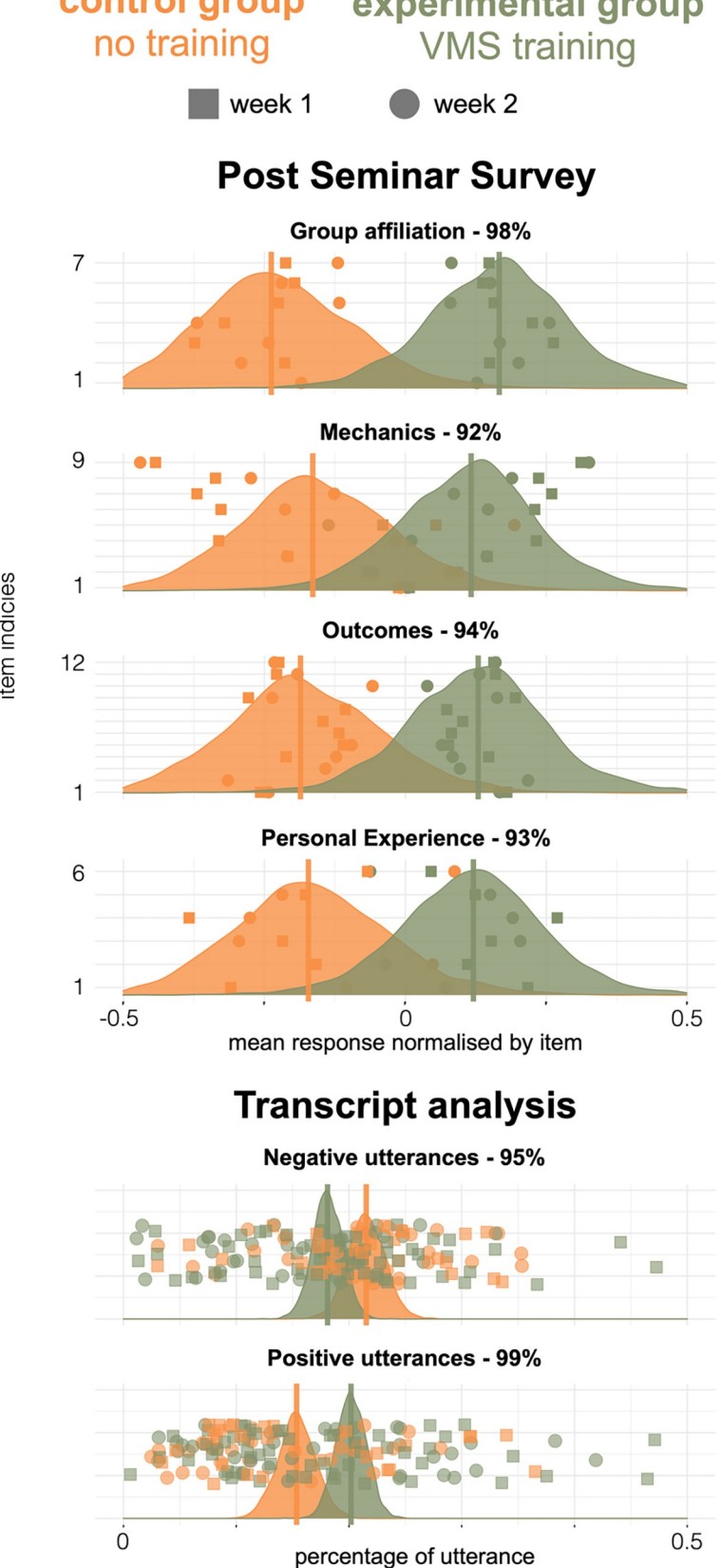

**Fig 2. Experiment 1 mean normalised responses for each survey item, of the proportion of utterances for each participant, in week 1 (squares) and week 2 (circles).** Average values for each survey theme or utterance emotion are shown by vertical lines and the probability distribution of those averages, as estimated by the models, are given shaded areas. The text of each item is given SM, by index. The probability that the training condition had an effect on participants' ratings (MPE) is given by the percentages above each plot.

We found very strong evidence that participants in the VMS training condition gave higher ratings than the control (MPE = 95%) across all survey items. Within each theme of the survey, the evidence was that they gave higher ratings to group affiliation (MPE = 98%), personal experience (MPE = 93%) and learning outcomes (MPE = 94%), and that the mechanics of their interaction were improved (MPE = 92%). We also found strong evidence that for the VMS trained group, there was a lower proportion of utterances that were negative (MPE = 95%), and a higher proportion that were positive (MPE = 99%).

**Discussion.** VMS training benefited our seminar students. They had a better personal experience, felt closer to their group, interacted with each other better, and thought that they had learnt more. Objective measures of their behaviour supported these ratings, showing that they used more positive and less negative language. We do not yet know why VMS improved the experience of our students. But we can speculate that there are three aspects of the training that—alone or in combination—could be responsible.

Firstly, it could be that the training session instills a set of *values*, implicitly or explicitly. During the training session, the students talk about the importance of committing to the process of a group meeting, and the value of interpersonal interaction. Regardless of the particular VMS techniques that are taught, it might be that simply discussing and endorsing these values positively shaped participants' interaction in the subsequent weeks.

Secondly, it could be that the *information* that the signals convey is beneficial. Knowing from their gestures that a certain number of people agree or disagree with a speaker's point, for example, is information that the group might not have without the VMS training. This increase in the common ground [16, 28–30] shared by the group could enhance their interaction, and have knock on positive effects for their learning and affiliation. If it is their informational content alone that gives the gestures their benefit, then the same advantage could be obtained by other means. Newer versions of the video conferencing platforms have implemented response buttons that superimpose an emoji on screen, such as round of applause that participants can use to indicate approval. As we test in Experiment 2, if these were used systematically, perhaps they could have an equal effect as VSM training.

Or perhaps not: finally, it could be precisely the physical *action* of the signals that is key to their benefit. One reason is that the signals are similar to iconic gestures such as thumbs up and waving that are probably already part of participants' gestural repertoire. And like real gestures, they can occur at the same time—and even be part of—the act of talking and listening [14]. As such they can be used fluidly and easily, compared to disengaging from the conversation to find the right button to trigger an applause emoji. An embodied gesture also has the potential to convey levels and nuances of a message, or add emotional tone, in a way that a emoji cannot [31, 32].

When two people mimic each other's' gestures and body language, it increases the social bond between them [18], and the ease with which they cooperate [33]. So another reason that the *action* of the signals may be crucial is that they induce similarity between the group members' movements. When a whole group coordinates their movement in this way, it increases the degree to which they are seen as an entity [34] and the affiliation they feel to each other [35–37]. We can hypothesise that perhaps it is these aspects of the action of an embodied gesture that meant we found strong effects on survey items and language that were related to the

emotional connection between participants. In the following experiment, we tested that hypothesis with a new experimental manipulation.

## Experiment 2

We pre-registered (https://osf.io/h79p6) a set of experiments with two goals. Firstly, we wanted to replicate the results of Experiment 1 with a different set of participants who were not students, who did not know each other previously, and who were engaged in a different range of group activities. Secondly, we wanted to contrast the *values*, *information* and *action* accounts of our results.

We achieved this with three experimental conditions. The *control* groups, consisted of a small groups of strangers that met online in Zoom, and were given a series of discussion activities. The *gesture* groups received a simplified version of the VMS training, shown to them in a short video at the start of the session. The *emoji* groups were shown a version of the same training, but the descriptions of particular gestures were replaced by instructions to use the response buttons for the same purposes.

We predicted that the *gesture* group would experience the same psychological benefits over the *control* group as we found in Experiment 1. As well as replicating our survey and transcript measures from Experiment 1, we sought to strengthen our results with standardised measures of social affiliation and interaction. The Networked Minds scale [38] is designed to measure the sense of social presence and togetherness during computer mediated interactions. The Empathic Concern (EC) and Perspective-Taking (PT) subscales of Interpersonal Reactivity Index [39] were used to assess the degree to which participants were aware of and attended to each others' thoughts and feelings.

We hypothesised that if the benefits of VMS were due to either the values instilled by the training, or the information conveyed by the gestures, then we would find that the *emoji* groups would also have improved experiences over the control groups. But if the psychological benefits of VMS depend, in part, on the physical *actions* of participants, then the gesture group would experience more psychological benefits than the *emoji* groups.

### Methods

**Participants.** 137 participants were recruited using the Prolific platform and remunerated at £10 the hour. Our criteria were that participants should be fluent English speakers residing in the UK, who had the computer equipment and internet connection necessary for video conferencing, and who were comfortable taking part in a recorded 'online focus group'. Sessions were open to a maximum of 8 participants, and were cancelled if fewer than 4 signed up or attended. 9 participants were excluded after the experiment because their internet connection dropped or they were unable to join the video conference, and 1 participant was dropped because they failed the instructional manipulation check. After exclusions, group sizes varied from 4 to 8, with a median of 6. There were 40 participants in the *control*, 41 in *emoji* and 46 in *gesture* groups. 84 participants were female, 30 were male, and 13 were non-binary or chose not to disclose. Ages range from 18 to 60 (M = 30.6, SD = 11.7).

Fully informed consent was obtained online, and participants' data were anonymised throughout. Participants were instructed to change their zoom screen name to a series of characters derived from their anonymised participant pool ID. Ethical approval was provided prior to data collection by the UCL board (3828/003, EP/2021/006).

**Procedure.** The experiment was listed with the title 'Covid-19 UK Lockdown Experience–Online Zoom Focus Group Study'. After giving their informed consent, participants were sent to the Gorilla platform. They picked a convenient time slot and filled in a series of

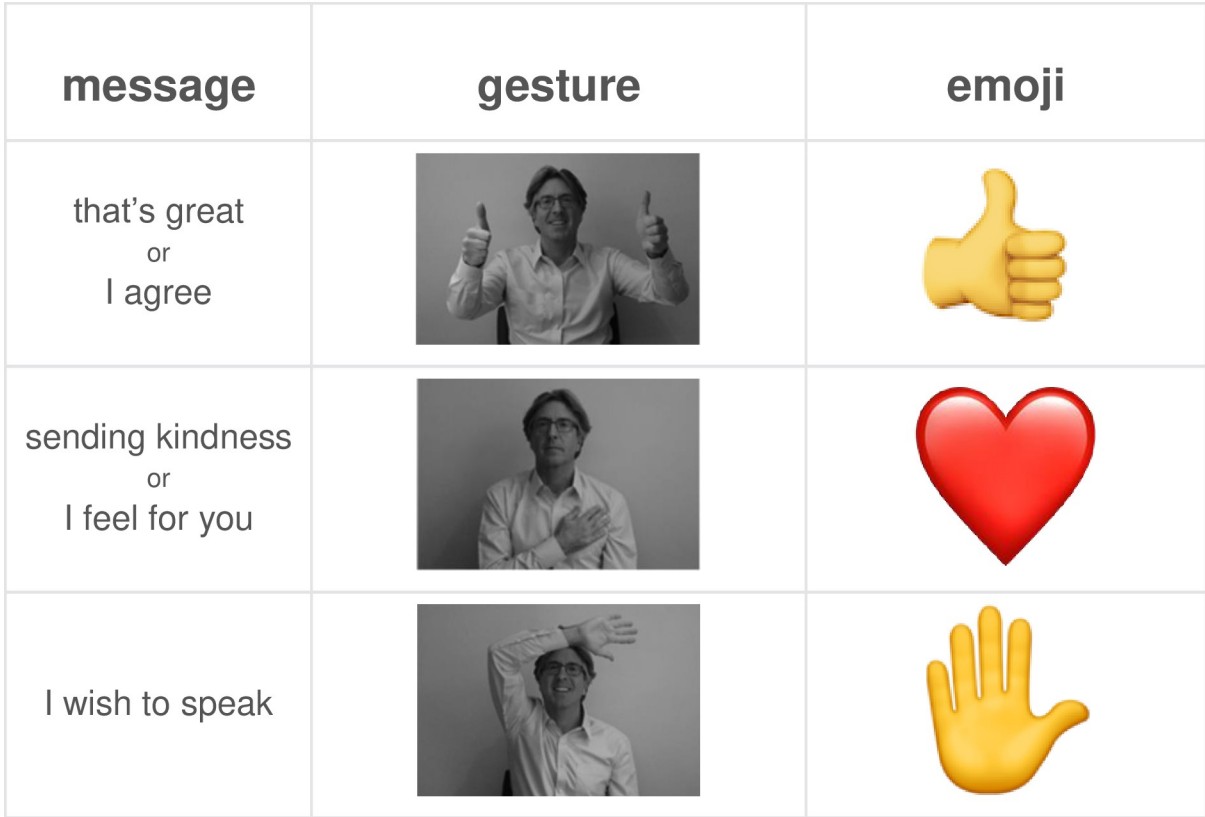

**Fig 3. Signals used in the *gesture* and *emoji* groups in Experiment 2.** Emoji's appeared superimposed in top right corner of participants' video.

demographic and personality measures, and a brief survey of their attitudes towards online meetings (these individual variables were collected for a Masters' project and are not reported in this manuscript).

At their chosen session time, the participants followed a zoom link. An experimenter, following a script, greeted them on arrival and reminded them to anonymise their zoom name if they had not done so. The experimenter explained that in the focus group participants would be discussing their experiences of the 2021 covid lockdown measures in the UK. While the *control* groups started their discussion session immediately, the *emoji* and *gesture* groups were first told they would be shown a 4 minute long video 'to give you some tools to facilitate online discussions'.

The videos instructed them to use a small set of signals during the discussion. For the gesture groups, they were asked to use just three of the VMS gestures from experiment 1. For the emoji group, they were told to use the equivalent Zoom response buttons (see Fig 3). Crucially, the two videos were identical when describing the meaning of a signal and when it should be used. They only differed when describing either the hand gesture or the emoji. After the videos, the experimenter asked the participants to practice performing a gesture or clicking to display an emoji. If any participants encountered difficulties, they were shown again how to do it. At no time were participants given the impression that they would be evaluating these tools. In both experimental groups, participants were given the impression that the main goal of the session was to share their experiences of lockdown with each other and the experimenter.

Then the participants began their discussion session. With participants' consent, the recording was started. The experimenter first told them to share as much as they would like to about their practical circumstances during lockdown. Discussions took around 10 mins.

Then the experimenter asked them to discuss amongst themselves 4 topics related to lockdown: *What surprised you*? *What was frustrating about living with the restrictions*? *What did you learn*? *What did you enjoy*? During the discussions, the experimenter turned off his video and microphone, so that he could not influence the group interactions.

After 15 mins the experimenter returned. Participants were thanked for their time, and instructed to follow a link to fill in a final survey on the Gorilla platform. First they were asked to recall the groups' shared experiences, and then they rated *How similar were the group members' experience of lockdown*? Then they completely a survey evaluating their experience of the discussion session. The survey used the same items as Experiment 1, but adapted them where necessary, for example, referring to the focus group and the discussion, rather than the seminar group and the learning goals. In addition, for this experiment we also used the standardised measures of the Networked Minds (NM) scale [38], and Empathic Concern (EC) and Perspective-Taking (PT) Subscales of Interpersonal Reactivity Index [39]. Finally, participants were debriefed and informed that a further goal of the experiment was to evaluate different tools to facilitate online interaction.

**Statistical analyses.** Following Experiment 1, we used Bayesian mixed models to quantify the evidence that each of our experimental conditions influenced participants' behaviour. S1 File give the explanatory power and full parameter estimates of each model with Median, Median Absolute Deviance (MAD), 95% Confidence-Interval (CI- CI+), Maximum Probability of Effect (MPE) and Overlap for each term.

Post discussion survey responses were reverse-scored where necessary and normalised within each item. The survey response were modelled as

response ~ condition * theme + (1 | group) + (1 | participant)

The NM scale was normalized and modelled as:

response ~ condition + (1 | group) + (1 | participant)

The EC and PT subscales of IRI were normalized and modelled as:

response ~ condition * scale + (1 | group) + (1 | participant)

The positive and negative utterances were modelled as

response ~ condition * valence + (1 | group) + (1 | participant)

In Fig 4 we show the observed data and the distribution of the means estimated by the model simulations. We report the maximum probability of effect (MPE)s below to quantify the probability that there are differences between pairs of condition means differ. MPE of above 90% s.

**Results.** Our results replicated the findings of Experiment 1 with different types of people engaged in a different type of online interaction with a highly reduced training. Non-student participants who were strangers to each other had an online discussion, and those that saw a brief VMS training video with gestures were more socially connected to each other. Compared to the control groups, they felt stronger group affiliation (MPE = 97%), demonstrated higher empathic concern (MPE = 93%), thought that they had more shared experiences (MPE = 93%) and scored higher on the Networked Minds scale (98%).

The groups that were trained to use response buttons and emojis had no such benefit from their training. There was no evidence that the emoji groups differed from the controls on any of the measures we used, apart from the fact that they did use more negative language (MPE = 99%) and less positive language (MPE = 99%). Comparing the two training groups with each other, it seems that the use of VMS gestures resulted in stronger affiliation (MPE = 98%), better perceived outcomes for the discussion (MPE = 91%) and a better

## Post Discussion Surveys

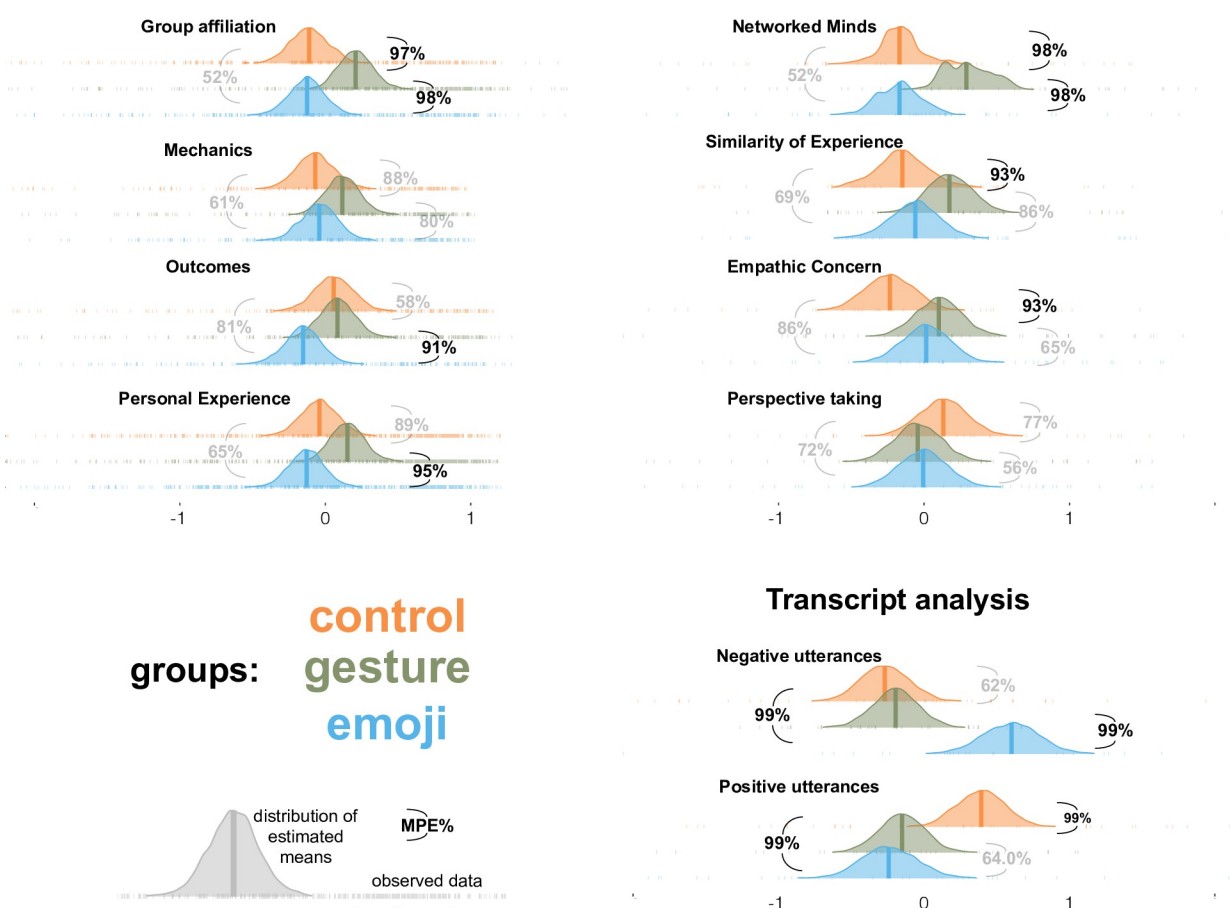

**Fig 4. Results of Experiment 2 post discussion surveys and transcript analysis.** Observed data are given in check marks, average values for each measure are shown by vertical lines, and the probability distribution of those averages as estimated by the models are given by shaded areas. The probability that there was a different any two conditions (MPE) is given in brackets, with those showing strong evidence in bold.

personal experience (MPE = 95%). Groups using the emojis also rated themselves lower on the networked minds scale (MPE = 98%), and used more negative language (MPE = 99%) than those that used gestures to achieve the same goals.

**Discussion.** We found, once more, that VMS training improved the experience of online social interactions. The results of Experiment 1 were replicated, and extended from students who knew each other talking about their classwork, to groups of more diverse strangers discussing their personal experiences. The results were strengthened by the use of standardised questionnaires, showing that groups with the VMS training scored higher on the Networked Minds and Empathic Concern scales.

Experiment 2 also gave us insight into *why* VMS training with gestures works. The emoji group watched a near identical training video, that also discussed how to signal agreement, empathy and a willingness to talk. The only difference was that the signal was an emoji rather than a hand gesture. But that difference resulted in less socially connected groups. Therefore, we can reject the idea that VMS training works because of the *values* conveyed by that—or any —training, and we can reject the idea that it is the *information* contained in the signal alone that has a positive benefit. Something about the use of gestures specifically appears to help

online interactions, whether it is the nuance of meaning they allow or the physical action of performing or seeing them performed by others.

There are interesting smaller patterns in the data of Experiment 2 that, arguably, did not fully replicate Experiment 1. Participants did not report that the mechanics of their interactions were better in the gesture groups compared to controls, or that the outcome of the discussion of their personal experience of it was better (though on both scales they were higher than the emoji groups). There were also more positive utterances in the control groups than either of the training groups. In general, we cannot tell if these minor differences were due to the fact that the VMS training in Experiment 2 was much reduced (a 4 minute video rather than a 45 minute interactive training session), or because the participants were different types of people (a broad range of the general public rather than students who knew each other), or because they were engaged in different types of activity (a discussion of personal experiences rather than academic classwork). These important questions will be addressed in future experimental contrasts and our analyses of how individual differences relate to the experience of VMS training and social interaction online.

## Conclusion

The results of two experiments were clear: short VMS training sessions produced measurable benefits for people in online social interaction. This is not an effect of having any training at all, and it is not solely the result of the information conveyed in the signal. Training in the use of response buttons to signal the same information with emojis did not produce the same benefits, and indeed in some cases seemed worse than no training at all. Quite why VMS gestures produce this effect, and possible limits to their use, is the subject of future studies. But we took the decision to publish this report now because of the potential benefits such training could have in the short term. Regardless of when lockdowns are fully lifted, it is clear that in the longer term, more of our working and studying lives will be spent in online meetings. VMS may provide a simple way to make those hours a little more productive and enjoyable.

## Supporting information

**S1 File.**
(PDF)

## Acknowledgments

We would like to thank the undergraduate students and first year tutors of the UCL Experimental Psychology Department for their participation, the University of Exeter Inclusivity Project, and Ben Hatcliffe for his assistance running the experimental groups online.

## Author Contributions

**Conceptualization:** Paul D. Hills, Mackenzie V. Q. Clavin, Miles R. A. Tufft, Matthias S. Gobel, Daniel C. Richardson.

**Data curation:** Miles R. A. Tufft, Daniel C. Richardson.

**Formal analysis:** Daniel C. Richardson.

**Funding acquisition:** Matthias S. Gobel.

**Investigation:** Paul D. Hills, Mackenzie V. Q. Clavin, Miles R. A. Tufft, Daniel C. Richardson.

**Methodology:** Paul D. Hills, Mackenzie V. Q. Clavin, Miles R. A. Tufft, Daniel C. Richardson.

**Project administration:** Paul D. Hills, Miles R. A. Tufft, Matthias S. Gobel, Daniel C. Richardson.

**Resources:** Paul D. Hills.

**Supervision:** Matthias S. Gobel, Daniel C. Richardson.

**Visualization:** Daniel C. Richardson.

**Writing – original draft:** Paul D. Hills, Mackenzie V. Q. Clavin, Miles R. A. Tufft, Daniel C. Richardson.

**Writing – review & editing:** Paul D. Hills, Mackenzie V. Q. Clavin, Miles R. A. Tufft, Matthias S. Gobel, Daniel C. Richardson.

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
