## [Decision Letter · Decision Letter 0]

12 Nov 2021

PONE-D-21-12225

Video Meeting Signals: A randomised controlled trial of a technique to improve the experience of video conferencing

PLOS ONE

Dear Dr. Richardson,

Thank you for submitting your manuscript to PLOS ONE. After careful consideration, we feel that it has merit but does not fully meet PLOS ONE’s publication criteria as it currently stands. Therefore, we invite you to submit a revised version of the manuscript that addresses the points raised during the review process.

It is requested to (see the enclosed comments):

-provide further details about VMS i.e. about the selection of gestures presented in figure1;

-describe the theoretical reference model adopted for the controlled trial to test efficacy of VMS;

-clarify how the themes included in the survey (group affiliation, the learning outcomes

of that seminar and the mechanics of the seminar interaction) were selected;

-clarify all variables of the trial: the manipulated/independent variable (e.g. VMS training or not), controlled variables, dependent variables (e.g. themes included in the survey);

-describe how were constructed the scales used for measurements of the themes;

- justify the apparent lack of a proper control condition;

- clarify if either group used the "emoticons" and/or the VMS gestures;

- specify if the authors also looked at the text-chat during the session and if this is included in the lexical analysis.

Moreover, a careful revision of the manuscript format is requested to improve the readability: the order of the sections of the manuscript ( e.g. Introduction, Method , Results, Discussion, Conclusions), the reference format as required by the journal.

We look forward to receiving your revised manuscript.

Kind regards,

Filomena Papa

Academic Editor

PLOS ONE

Additional Editor Comments (if provided):

The manuscript needs some clarifications. More details about how the VMS gestures were selected should be provided to the readers.The theoretical reference model adopted in the design the controlled trial and the criteria adopted for selection of the themes included in the survey (group affiliation, the learning outcomes of that seminar and the mechanics of the seminar interaction) need to be described including the appropriate references. The controlled trial design needs to be clarified identifying all trial variables: the manipulated/independent variable (e.g. VMS training or not), controlled variables, dependent variables (e.g. themes included in the survey). The procedure used for construction of scales used for measurement of the themes has to be described mentioning the relevant references.

Finally, the format of the manuscript has to be reviewed to improve readability and according to the format required by the journal (e. g. order of sections, references).

Journal Requirements:

6. We note that Figure 1 includes an image of a participant in the study. 

Reviewers' comments:

Reviewer's Responses to Questions

**Comments to the Author**

1. Is the manuscript technically sound, and do the data support the conclusions?

Reviewer #1: Yes

Reviewer #2: Partly

2. Has the statistical analysis been performed appropriately and rigorously? 

Reviewer #1: Yes

Reviewer #2: I Don't Know

3. Have the authors made all data underlying the findings in their manuscript fully available?

Reviewer #1: Yes

Reviewer #2: Yes

4. Is the manuscript presented in an intelligible fashion and written in standard English?

Reviewer #1: Yes

Reviewer #2: Yes

5. Review Comments to the Author

Reviewer #1: In the manuscript, "Video Meeting Signals: A Randomized Control Trial of a Technique to Improve the Experience of Video Conferencing," the authors trained a group of participants on the VMS(tm) system or provided no training. Participants assigned to the VMS group learned a set of (9) gestures to implement during their online Zoom classes. Participants interactions were annotated and transcribed in Zoom, and participants were asked to complete a post-seminar survey at the end of each class session. Survey questions were grouped into four themes (i.e., group affiliation, learning outcomes, personal experience, seminar mechanics). Two analyses were conducted (1) survey response data and (2) linguistic content from transcripts. Findings suggested that participants assigned to the VMS training had an overall better experience in the Zoom meeting that participants not assigned to the training group - both in the survey and more positively valanced language used in the transcript.

Overall, the manuscript was easy to read, clear, and to the point. It seems, based on the results from the data, the VMS system has a positive impact on participant performance. As the authors suggest, they are not able to extrapolate why the VMS system works, but they argue that implementing this type of gestural system in video meetings might be very helpful. I am inclined to agree, but I have a few questions and suggestions for the authors to add to the manuscript to clarify the importance and impact of such a gestural system.

Though the authors show clear differences in the two groups, it is not clear if either group used the "emoticons" and/or the VMS gestures. Is it possible for the authors to descriptively provide this data? Adding the frequency at which emoticons are used in the Zoom session and the rate at which gestures naturally (and post-training) occurred in either setting is also needed to fully understand the impact of the training. Also, did the authors also look at the text-chat during the session and is this included in the lexical analysis?

The finding is simple and preliminary, but I appreciate the authors mention of this, as to not oversell the product. Other than the request for extra detail on the use of the emoticons and gestures, I have no major concerns for the manuscript.

Reviewer #2: The manuscript reports a study in which over 100 students were trained with gestural signals for use in online seminar sessions taking place over two consecutive weeks. The students then completed several questionnaires evaluating the rapport with their seminar group, their personal experience, learning outcomes, the mechanics if their interactions, as well as the valence of utterances they made during the interactions. The group was compared on these measures to a group that did not receive the gestural signaling training. The measures showed that the group who received the training scored higher on the questionnaire measures and that the valence of their utterances were more positive.

The reported study addresses a research question of great significance in current times, namely how online/video-communication platforms like Zoom can be improved to allow for more effective and pleasant interactions. There are several things I like about this study, but also some points that give me concern.

Positive points are the number of participants, which is a decent sample size for meaningful analyses of psychometric/social questionnaire measures. Also positive is the use of advanced statistics, including Bayesian techniques. The manuscript is clearly written and easy to read.

My main point of concern is the apparent lack of a proper control condition. Unless I missed it or misunderstood, it sounds like the control group just received no treatment at all. If this is indeed the case, then any effects measured may be attributable to a pure placebo effect. Such an effect could emerge from the control group’s knowledge that they did receive some form of training that was meant to influence interactions, or even simply because they had experienced an additional, in-depth social interaction session for the training. The question is if the control group had received some other treatment that also involved 60 minutes of in-depth social interaction but with a focus on practicing something entirely different, would the same effects have been observed? Or, even if something different had been taught that was introduced as meaning to benefit interaction (e.g. how to best set the camera angle and sound levels, adjust background etc. to allegedly make interaction most effective), would this have had a comparable effect? Without a proper control treatment, it isn’t possible to attribute the effects found to the effectiveness of gesture training per se. Unfortunately, this makes it very difficult to draw meaningful conclusions from this study that are specific to the benefit of the VSM training (over and above any other in-depth group interaction). If the authors disagree I’d be interested to hear their arguments.

6. PLOS authors have the option to publish the peer review history of their article (what does this mean?). If published, this will include your full peer review and any attached files.

Reviewer #1: No

Reviewer #2: No

---

## [Author Response · Author response to Decision Letter 0]

24 Jan 2022

Please find below a point by point reply to each comment made in the review process. As you will see, we have now included a second substantial pre-registered experiment, that we believe answer each of the valid concerns that were raised about our first submission. 

Editor Comments

The manuscript needs some clarifications. More details about how the VMS gestures were selected should be provided to the readers.

As we have now explained in the MS, this set of gestures were developed over time by PDH through working with groups online. 

The theoretical reference model adopted in the design the controlled trial and the criteria adopted for selection of the themes included in the survey (group affiliation, the learning outcomes of that seminar and the mechanics of the seminar interaction) need to be described including the appropriate references. 

The controlled trial design needs to be clarified identifying all trial variables: the manipulated/independent variable (e.g. VMS training or not), controlled variables, dependent variables (e.g. themes included in the survey). The procedure used for construction of scales used for measurement of the themes has to be described mentioning the relevant references.

As we now state in the MS, we used a cluster randomised trial. All trial variables are stated in full in the MS and/or the supplementary material. We specify the statistical models used in our analysis in the main text, and explain in more detail in the supplementary materials. For experiment 1, we could not find a validated scale that was adequate for our use. For this reason, the survey questions were written by us, and grouped into intuitive themes. we were sure to be transparent about our data. In the MS we show the mean responses to individual items, as well as analysing by themes, and the full data set is available online. For the new experiment 2 in our resubmission, we also included two validated measures of social affiliation and interaction. 

Finally, the format of the manuscript has to be reviewed to improve readability and according to the format required by the journal (e. g. order of sections, references).

We have restructured the MS as requested.

*In the ethics statement in the Methods and online submission information, please ensure that you have specified what type you obtained

Consent was given online in the initial survey completed before the first survey. This has been described in the MS.

*We note that Figure 1 includes an image of a participant in the study.

Actually, it does not: that is one of the paper authors (P.H.) who is shown teaching the signals. We have made this clear in the figure legend. 

*Reviewer #1: 

*Overall, the manuscript was easy to read, clear, and to the point. It seems, based on the results from the data, the VMS system has a positive impact on participant performance. As the authors suggest, they are not able to extrapolate why the VMS system works, but they argue that implementing this type of gestural system in video meetings might be very helpful. I am inclined to agree, but I have a few questions and suggestions for the authors to add to the manuscript to clarify the importance and impact of such a gestural system.

We thank R1 for their kind assessment of the MS and its strengths.

*Though the authors show clear differences in the two groups, it is not clear if either group used the "emoticons" and/or the VMS gestures. Is it possible for the authors to descriptively provide this data? Adding the frequency at which emoticons are used in the Zoom session and the rate at which gestures naturally (and post-training) occurred in either setting is also needed to fully understand the impact of the training. Also, did the authors also look at the text-chat during the session and is this included in the lexical analysis?

We are happy to clarify this important point: during the zoom sessions emoticons were not used at all. Indeed, the emoticons that are currently available in zoom (and were employed in our new experiment 2) were not implemented in the zoom platform at the time of the experiment. And so we can be confident they were not used. The text chat was only used in a very limited way to exchange to exchange links and information. In the MS we make these important points clear to the reader.

*The finding is simple and preliminary, but I appreciate the authors mention of this, as to not oversell the product. Other than the request for extra detail on the use of the emoticons and gestures, I have no major concerns for the manuscript.

We are delighted with the reviewer’s assessment, and hope that the addition of a new, larger experiment that replicates and extends our findings will be welcome. 

*Reviewer #2:

*The reported study addresses a research question of great significance in current times, namely how online/video-communication platforms like Zoom can be improved to allow for more effective and pleasant interactions. There are several things I like about this study, but also some points that give me concern.

We believe that the new experimental data we have now collected and included in this MS convincingly answer the questions that they raise. 

*Positive points are the number of participants, which is a decent sample size for meaningful analyses of psychometric/social questionnaire measures. Also positive is the use of advanced statistics, including Bayesian techniques. The manuscript is clearly written and easy to read.

We thank the review for this kind assessment of the MS’s strengths.

*My main point of concern is the apparent lack of a proper control condition. Unless I missed it or misunderstood, it sounds like the control group just received no treatment at all. If this is indeed the case, then any effects measured may be attributable to a pure placebo effect. Such an effect could emerge from the control group’s knowledge that they did receive some form of training that was meant to influence interactions, or even simply because they had experienced an additional, in-depth social interaction session for the training. The question is if the control group had received some other treatment that also involved 60 minutes of in-depth social interaction but with a focus on practicing something entirely different, would the same effects have been observed?

* Or, even if something different had been taught that was introduced as meaning to benefit interaction (e.g. how to best set the camera angle and sound levels, adjust background etc. to allegedly make interaction most effective), would this have had a comparable effect? Without a proper control treatment, it isn’t possible to attribute the effects found to the effectiveness of gesture training per se. Unfortunately, this makes it very difficult to draw meaningful conclusions from this study that are specific to the benefit of the VSM training (over and above any other in-depth group interaction). If the authors disagree I’d be interested to hear their arguments.

We agree completely with the reviewers assessment here. In our discussion we described three key differences between our experimental and control conditions, all of which - including what R2 terms the placebo effect - could be the causal reason for the differences we see. 

For this reason, when submitting the MS we pre-registered a set of experiments that could draw more precise conclusions about the benefits of the VSM training. We have now completed one of those studies, and report in the revised manuscript as Experiment 2. We believe this substantially increases the scope and impact of this paper, because we now have strong evidence that precisely addresses the reviewer’s only point of concern. 

In Experiment 2, new groups of participants had an hour long zoom session. In the VMS condition, they received a 4 minute video instructed them to use of VMS gestures. In the emoji condition, they received a training video that was identical, except that they were told to use zoom reaction buttons to display emojis on their video feeds. In the control condition, they were given no training. As in experiment 1, after the session participants evaluated their interaction and reported their feelings. There is strong evidence that participants in the symbol condition had a worse experience that those in the VMS gesture condition. 

We can confidently reject with evidence, therefore, the possibility that R2 raises: that we would get the same benefit with any sort of pre-interaction training. This new experimental contrast allows us to pin the benefits of VMS to the particular use of gesture. When people are trained to signal the same information at the same time with the same training - but to use symbols - there is no advantage. We are confident that R2 will agree that this new data allows for a meaningful conclusion to be drawn about VMS and substantially increases the value of this paper.

---

## [Decision Letter · Decision Letter 1]

25 Mar 2022

PONE-D-21-12225R1

Video Meeting Signals: Experimental evidence for a technique to improve the experience of video conferencing

PLOS ONE

Dear Dr. Richardson,

Thank you for submitting your manuscript to PLOS ONE. After careful consideration, we feel that it has merit but does not fully meet PLOS ONE’s publication criteria as it currently stands. Therefore, we invite you to submit a revised version of the manuscript that addresses the points raised during the review process.

It is requested to revise the manuscript according to Reviewer 2's comments (see after).

We look forward to receiving your revised manuscript.

Kind regards,

Filomena Papa

Academic Editor

PLOS ONE

Journal Requirements:

Additional Editor Comments (if provided):

The manuscript improved a lot after revision due to the inclusion of Experiment 2.

Carefully revise the manuscript to correct typos (for instance on page 5 substitute "Statistical analyses:" with "Statistical analyses").

Reviewers' comments:

Reviewer's Responses to Questions

**Comments to the Author**

1. If the authors have adequately addressed your comments raised in a previous round of review and you feel that this manuscript is now acceptable for publication, you may indicate that here to bypass the “Comments to the Author” section, enter your conflict of interest statement in the “Confidential to Editor” section, and submit your "Accept" recommendation.

Reviewer #1: All comments have been addressed

Reviewer #2: (No Response)

2. Is the manuscript technically sound, and do the data support the conclusions?

Reviewer #1: Yes

Reviewer #2: Yes

3. Has the statistical analysis been performed appropriately and rigorously? 

Reviewer #1: Yes

Reviewer #2: Yes

4. Have the authors made all data underlying the findings in their manuscript fully available?

Reviewer #1: Yes

Reviewer #2: Yes

5. Is the manuscript presented in an intelligible fashion and written in standard English?

Reviewer #1: Yes

Reviewer #2: Yes

6. Review Comments to the Author

Reviewer #1: I feel that the authors have more than satisfactorily responded to my comments, and the comments of the other reviewer.

Reviewer #2: I have read the revised manuscript and think the authors have done a great job at addressing the different concerns raised. I am particularly pleased with them including Experiment 2, which indeed does address my main concern.

I have two final requests though:

1) For experiment 2, rather than referring to the SMs, please include the full model specifications into the statistical analysis section, as was done for Experiment 1. The statistical model applied is a core part of the study and therefore should not be ‘hidden’ in the SMs/more effortful to consult than any of the other details.

2) One thing I am really missing regarding both experiments is an overview of how much people actually made use of the signals (i.e. gestures and emojis). This is important, because it gives insight into whether the effects are purely down to having the opportunity to draw on the respective signals, or whether actually using them plays a role (I am sure they made good use of them, but the reader should be able to validate this). I don’t expect the authors to add much here in terms of analysis, but summary statistics (e.g., means, sd, min max values) for the different conditions and a comparison between them, minimally, should be included to allow the reader to fully evaluate the findings (especially the difference between the emoji and gesture effects….which in theory could be due to a difference in how much people used the emojis versus the gestures).

Editorial details:

Except for in the headings/titles, the font is too small in figures 2 and 4.

Typos:

p. 13: Post discussion survey responses were reversed-scored where necessary and normalised within each item.

 reverse-scored

p. 15 The only difference was that the signal was as emoji rather than a hand gesture.

 an emoji

p. 15: There were also more positive utterances in the control groups than either of the trainman groups.

 training?

7. PLOS authors have the option to publish the peer review history of their article (what does this mean?). If published, this will include your full peer review and any attached files.

Reviewer #1: No

Reviewer #2: No

---

## [Author Response · Author response to Decision Letter 1]

2 Jun 2022

Dear Editor,

Here we confirmed, point by point, where we have acted upon the final comments. 

"The manuscript improved a lot after revision due to the inclusion of Experiment 2. Carefully revise the manuscript to correct typos (for instance on page 5 substitute "Statistical analyses:" with "Statistical analyses”)."

Done  

"Reviewer #1: I feel that the authors have more than satisfactorily responded to my comments, and the comments of the other reviewer.

Reviewer #2: I have read the revised manuscript and think the authors have done a great job at addressing the different concerns raised. I am particularly pleased with them including Experiment 2, which indeed does address my main concern."

We are delighted that both reviewers agree that our revisions satisfactorily resolved the comments in their reviews. 

"1) For experiment 2, rather than referring to the SMs, please include the full model specifications into the statistical analysis section, as was done for Experiment 1. The statistical model applied is a core part of the study and therefore should not be ‘hidden’ in the SMs/more effortful to consult than any of the other details. "

We have moved the model specifications from the SM to the main text.

"2) One thing I am really missing regarding both experiments is an overview of how much people actually made use of the signals (i.e. gestures and emojis). This is important, because it gives insight into whether the effects are purely down to having the opportunity to draw on the respective signals, or whether actually using them plays a role (I am sure they made good use of them, but the reader should be able to validate this). I don’t expect the authors to add much here in terms of analysis, but summary statistics (e.g., means, sd, min max values) for the different conditions and a comparison between them, minimally, should be included to allow the reader to fully evaluate the findings (especially the difference between the emoji and gesture effects….which in theory could be due to a difference in how much people used the emojis versus the gestures)."

Unfortunately, we do not have access to this information in our data. The zoom video recording and transcripts do not code for the use of emojis (or at least, they did not in the version of the software that we employed). Therefore we cannot supply this information to the readers. Crucially, however, this information does not impact the claims that we make in the paper. 

Firstly, there might be concern that after the training, participants did not use either gestures or emojis *at all*. In both the gesture and the emoji conditions, participants are trained in their use. The Experimenter asks then to repeat back the signs and practice them, confirming that they are able to use them. And as we discuss in the MS, participants did indeed use the signals as instructed (according to the Experimenter’s anecdotal observations at the time). Most importantly, there are rich differences in participants’ experiences and behaviour between our no training, gesture and emoji conditions, as our data show. So does not appear to be the case that participants ignored or failed to engage with the signalling aspect of the experiment. 

Secondly, though they can and did use gestures and emojis, there might be a concern that there was a difference in the frequency of usage between conditions. If participants then choose not to use either the emojis or the gestures, or use one less than the other, then presumably that is because of something intrinsic to those signals: for example, they are less convenient to use, they aren’t as expressive, they don’t help communication. It is precisely those aspects of the use of signals that concerns us, and that we discuss in our results. To repeat: a possible difference in the frequency of usage between conditions does not change the content or evidential strength of any of our claims.

"  Except for in the headings/titles, the font is too small in figures 2 and 4."

We have used the font size of the headings as the minimum font size throughout. Unfortunately, this meant that we could not include the abbreviated text of each survey item in figure 4. These now refer by index to the full text in the SM. 

" Typos: p. 13: Post discussion survey responses were reversed-scored where necessary and normalised within each item.  reverse-scored p. 15 The only difference was that the signal was as emoji rather than a hand gesture.  an emoji p. 15: There were also more positive utterances in the control groups than either of the trainman groups.  training?"

All these typos have been fixed.

---

## [Editor Report · Decision Letter 2]

10 Jun 2022

Video Meeting Signals: Experimental evidence for a technique to improve the experience of video conferencing

PONE-D-21-12225R2

Dear Dr. Richardson,

We’re pleased to inform you that your manuscript has been judged scientifically suitable for publication and will be formally accepted for publication once it meets all outstanding technical requirements.

Kind regards,

Emily Chenette

Editor in Chief

PLOS ONE
---

## [Editor Report · Acceptance letter]

7 Jul 2022

PONE-D-21-12225R2 

Video Meeting Signals: Experimental evidence for a technique to improve the experience of video conferencing 

Dear Dr. Richardson:

I'm pleased to inform you that your manuscript has been deemed suitable for publication in PLOS ONE. Congratulations! Your manuscript is now with our production department. 

Kind regards, 

on behalf of

Dr Emily Chenette 

Staff Editor

PLOS ONE